# Albumin Urinary Excretion Is Associated with Increased Levels of Urinary Chemokines, Cytokines, and Growth Factors Levels in Humans

**DOI:** 10.3390/biom11030396

**Published:** 2021-03-08

**Authors:** Bengt Fellström, Johanna Helmersson-Karlqvist, Lars Lind, Inga Soveri, Måns Thulin, Johan Ärnlöv, Kim Kultima, Anders Larsson

**Affiliations:** 1Department of Medical Sciences, Uppsala University, Uppsala University Hospital, SE-751 85 Uppsala, Sweden; bengt.fellstrom@medsci.uu.se (B.F.); johanna.helmersson.karlqvist@akademiska.se (J.H.-K.); lars.lind@medsci.uu.se (L.L.); Inga.soveri@medsci.uu.se (I.S.); kim.kultima@medsci.uu.se (K.K.); 2Centre for Statistics, The Swedish University of Agricultural Sciences, SE-752 36 Uppsala, Sweden; mans@statistikkonsult.com; 3Division of Family Medicine and Primary Care, Department of Neurobiology, Care Sciences and Society (NVS), Karolinska Institutet, 14152 Huddinge, Sweden; johan.arnlov@ki.se

**Keywords:** glomerular injury, urine, thrombospondin 2, interleukin 6, interleukin 8, hepatocyte growth factor

## Abstract

The aim of the present study was to study the associations between urine albumin excretion, and a large number of urinary chemokines, cytokines, and growth factors in a normal population. We selected 90 urine samples from individuals without CVD, diabetes, stroke or kidney disease belonging to the Prospective Investigation of the Vasculature in Uppsala Seniors Study (41 males and 49 females, all aged 75 years). Urinary cytokine levels were analyzed with two multiplex assays (proximity extension assays) and the cytokine levels were correlated with urine albumin. After adjustment for sex, body mass index (BMI), estimated glomerular filtration rate (eGFR), smoking and multiplicity testing, 11 biomarkers remained significantly associated with urine albumin: thrombospondin 2, interleukin 6, interleukin 8, hepatocyte growth factor, matrix metalloproteinase-12 (MMP-12), C-X-C motif chemokine 9, tumor necrosis factor receptor superfamily member 11B, osteoprotegerin, growth-regulated alpha protein, C-X-C motif chemokine 6, oncostatin-M (OSM) and fatty acid-binding protein, intestinal, despite large differences in molecular weights. In this study, we found associations between urinary albumin and both small and large urine proteins. Additional studies are warranted to identify cytokine patterns and potential progression markers in various renal diseases.

## 1. Introduction

The measurements of urinary albumin/total protein excretion play key roles in the detection and classification of renal disease [1]. Even small amounts of albumin, i.e., microalbuminuria, are associated with increased morbidity and mortality and are therefore part of the chronic kidney disease staging according to the Kidney Disease: Improving Global Out- comes (KIDGO) guidelines [2].

Water and small molecules can freely pass through the glomerular filter. For larger molecules, the permeability is depending on the size. Under normal conditions the glomerular membrane is almost impermeable to albumin due to the size and charge of albumin [3,4].

The reference method for albuminuria is the measurement of albumin in a 24-h urine collection. Twenty-four-hour collections are burdensome for patients to perform and subject to errors resulting from erroneous collection periods or incomplete collections. Thus, the 24-h collections have usually been replaced by spot urine samples collected in the morning or second void. The amounts of albumin in spot samples are usually expressed as mg/L or creatinine adjusted as mg albumin/g creatinine. The creatinine adjustments are used to correct for urine dilutions.

Considering the importance of albuminuria for morbidity and mortality [5], we wanted to investigate the associations between albuminuria and urine cytokine levels. Albuminuria is a marker for glomerular damage and tissue damage is associated with inflammatory response. Thus, it seems likely that albuminuria could be associated with urinary inflammatory cytokine levels especially if there are active inflammatory processes. In the present study, we focused on study subjects with relatively normal glomerular filtration rates (GFR) to minimize the effects of prerenal accumulation of cytokines secondary to reduced GFR.

The aim of the present study was to study the associations between glomerular damage, defined as urine albumin excretion, and a large number of urinary chemokines, cytokines, and growth factors in a cohort of elderly individuals without CVD, diabetes or stroke. We used a combination of two multiplex proximity extension assays (PEAs) to detect more than 150 chemokines, cytokines, and growth factors in the urine samples.

## 2. Materials and Methods

### 2.1. Patients and Methods

The study subjects were part of the Prospective Investigation of the Vasculature in Uppsala Seniors (PIVUS) study cohort [6]. The subjects were chosen from persons living in the community of Uppsala, Sweden. The urine samples for this study were collected at the age of 75 years. All participants were medically examined and given a questionnaire regarding their medical history, lifestyle, including smoking and exercise habits [7]. A total of 90 individuals without CVD, diabetes, or stroke were included. The samples were stored at <−70C prior to the analysis. The study was conducted in accordance with the Declaration of Helsinki. All participants in the study provided written informed consent, and the Ethics Committee of Uppsala University approved the study protocol.

### 2.2. Urine Albumin and Creatinine Determinations

Urine albumin was measured using particle enhanced immunological reagents from Abbott Laboratories (Abbott Park, IL, USA) on an Architect Ci8200 analyzer (Abbot Laboratories, Abbott Park, IL, USA). Urinary creatinine was analyzed with an IDMS-calibrated modified kinetic Jaffe reaction also on the Architect Ci8200 analyzer (Abbott Laboratories, Abbott Park, IL, USA). A urine albumin of >20 mg/L or a urine albumin/creatinine ratio of >2.5 mg/mmol is suggestive of microalbuminuria.

### 2.3. Determination of Urinary Chemokines, Cytokines, and Growth Factors

The urinary chemokines, cytokines, and growth factors were analyzed using Proseek Multiplex Cardiovascular II and Inflammation I panels (Olink Bioscience, Uppsala, Sweden). The two panels were analyzed in separate plates [8,9]. Briefly, 1 µL urine diluted 1:4 in saline was mixed with 3 µL incubation mix containing paired antibodies labeled with unique corresponding DNA oligonucleotides. The mixture was first incubated overnight at 8 °C. Then, 96 µL extension mix containing PEA enzyme and PCR reagents was added, and the samples were incubated for 5 min at room temperature before the plate was transferred to a thermal cycler for 17 cycles of DNA amplification. A 96.96 Dynamic Array IFC (Fluidigm, South San Francisco, CA, USA) was prepared and primed according to the manufacturer’s instructions. In a separate plate, 2.8 µL of sample mixture was mixed with 7.2 µL detection mix from which 5 µL was loaded into the right side of the primed 96.96 Dynamic Array IFC. The unique primer pairs for each cytokine were loaded into the left side of the 96.96 Dynamic Array IFC, and the protein expression program was run in Fluidigm Biomark reader according to the instructions for Proseek. Each of the Proseek panels analyzed 92 biomarkers, a total of 194 biomarkers.

### 2.4. Statistics

Cytokine values above or below the highest and lowest standard points were assigned the values of these points. Protein levels were measured on a log2-scale and further transformed to an SD-scale in order to be easily comparable. Linear regression analysis was applied to relate each urinary protein (dependent variable) vs. urinary albumin (independent variable). Each urinary protein was evaluated in a separate model. Urinary creatinine, sex (age same in all subjects), BMI, current smoking and eGFR were used as confounders in all models (additional independent variables). All the continuous independent variables, except BMI, were ln-transformed. Analyzing a large number of relationships increases the risk of false positive findings. The *p*-values were therefore adjusted for multiplicity using the false discovery rate (FDR) approach with 0.05 as the limit [10]. STATA 16 (Stata Inc, College Station, TX, USA) was used for the calculations.

## 3. Results

### 3.1. Study Cohort

The cohort consisted of 41 males and 49 females. The basic characteristics of the study cohort are presented in Table 1. Of the investigated cytokines only intestinal fatty acid-binding protein with a molecular weight of 15 kDa (*p* = 0.000028) and Cystatin D with a molecular weight of 16.1 kDa (*p* = 0.0004713) were significantly associated with eGFR according to linear regression.

### 3.2. Significant Associations between Urine Cytokines and Urine Albumin

After adjustment for sex, urinary creatinine, eGFR, BMI, smoking, exercise habits and multiplicity testing, 11 biomarkers remained significantly associated with urine albumin (thrombospondin 2, interleukin 6, interleukin 8, hepatocyte growth factor, matrix metalloproteinase-12 (MMP-12), C-X-C motif chemokine 9, tumor necrosis factor receptor superfamily member 11B, osteoprotegerin, growth-regulated alpha protein, C-X-C motif chemokine 6, oncostatin-M (OSM) and fatty acid-binding protein, intestinal) (Table 2).

### 3.3. The Influence of Molecular Weights on the Associations between Urine CYtokines and Urine Albumin

The molecular weights of the cytokines, showing significant associations with urine albumin excretion, varied between 10.2 and 130 kDa which includes proteins above and below the glomerular threshold for filtration of proteins. According to Spearman Rank correlation there were no clear influences of the molecular weight on the associations between urine cytokines and urine albumin.

## 4. Discussion

The study subjects were recruited from a community living cohort without any prior diagnosed kidney injury or compromised eGFR. We showed that there were strong associations between the urine albumin measured in mg/L and several urinary cytokines, chemokines, and growth factors. The study subjects were elderly individuals, probably suffering from some degree of age-related renal dysfunction, including (to some extent) nephrosclerosis. However, they do not report any specific renal disease. Being an elderly group of individuals, they are representative of a normal population that most frequently seek health care.

The main role of kidney glomeruli is to retain large macromolecules while allowing the transfer of small waste products into the urine. The glomerular filter is a size- and charge-selective barrier that allows very low amounts of albumin (molecular weight 67 kDa) to be transferred into the urine. Proteins with molecular weights below approximately 35 kDa are not retained by the glomeruli but are freely filtered into the primary urine. Theoretically, an increase in urine albumin caused by glomerular damage should be associated with an increase in other large plasma proteins. In this study, we found associations between urine albumin concentrations and both small and large urine proteins. The increases are thus not caused by a strict increased leakage of large plasma proteins. This indicates that several of the proteins may be produced locally in the kidney. This is further supported by the limited number of biomarkers that were significantly associated with eGFR and by previous publications showing the direct roles of the significantly associated markers in kidney diseases.

Albuminuria is not only used as a marker of renal dysfunctions but it is also a well-recognized predictor of cardiovascular risk in both community-based populations and high-risk populations. In several studies, proinflammatory cytokines have been shown to be cardiovascular risk markers. Acute or chronic injury to the kidneys or heart is known to be able to induce acute or chronic dysfunction of the other organ. These interactions are often referred to as the cardiorenal syndrome. The mechanisms behind these interactions are poorly understood but could be mediated by cytokines. The aim of this study was exploratory to investigate if any of these markers do appear in urine from apparently healthy people. The results indicate that the urine cytokine levels were not due to filtration in the glomeruli.

Thrombospondin-2 has been shown to act as a regulator of inflammation and matrix remodeling in experimental kidney disease and thrombospondin-2 gene therapy worsened the nephropathy [11]. Interleukin 6 [12] and CXCL1 [13] have been suggested to be involved in the progression of renal injury in IgA nephritis, while IL-8 levels were associated with poor prognosis in idiopathic membranous nephropathy [14]. Human HGF treatment of normal rats caused proteinuria [15]. Both MMP-12 [16] and C-X-C motif chemokine 9 [17] have been shown to contribute to glomerular injury in anti-glomerular basement membrane glomerulonephritis. Oncostatin-M participated in immune regulation and lack of oncostatin-M resulted in increased levels of anti-dsDNA and glomerulonephritis [18].

Urinary levels of Osteoprotegerin were positively associated with renal involvement in lupus nephritis patients [19]. Intestinal fatty acid-binding protein has been reported to be a promising biomarker for diabetic nephropathy [20]. Thus, the urine biomarkers that we found to be associated with the glomerular injury marker urine albumin have previously been reported to be associated with different types of kidney injury and diseases. The study subjects were recruited from a community living cohort without any prior diagnosed kidney injury or compromised eGFR. Apparently healthy people do excrete multiple biomarkers captured with this proteomics platform, in addition to albumin. These results may serve as a reference when studying patients with specific renal diseases.

Some limitations of the study should be noted, such as targeting a normal population consisting of 75-year-old individuals who may have a certain degree of nephrosclerosis and a slightly reduced renal function. Urinary albumin is not a pure and exclusive marker of increased glomerular permeability, since some albumin is filtered in glomeruli and subsequently reabsorbed in the proximal tubule.

The determination of broad urine cytokine panels is a new concept and there are therefore no internationally accepted calibrators for most of the studied biomarkers and no well-established reference values. The lack of international calibrators means that each company has to develop their own calibrations which makes it difficult to compare results obtained with assays from different manufacturers.

In conclusion, the associations in this study were present in a normal population and could be used as comparator in future studies in specific kidney diseases. Further studies are warranted to investigate if the differential urine patterns could be identified in various kidney diseases and if any of these cytokines or cytokine patterns could be used to predict progression of renal dysfunction or response to treatment in renal disease.

## Figures and Tables

**Table 1 biomolecules-11-00396-t001:** Basic characteristics of the population (n = 90). eGFR = estimated glomerular filtration rate.

Variable	Number/Mean	Percent/Standard Deviation (SD)
Age (years)	75	
Males/females	41/49	45.6/54.4%
Smoker		6.66%
	**Mean**	**SD**
Body mass index (BMI)	26.9	3.79
eGFR_combo_ (mL/min/1.73 m^2^)	70.3	12.4
Urine albumin (mg/L)	16.5	28.2

**Table 2 biomolecules-11-00396-t002:** Relationships between 79 urinary proteins and urinary albumin adjusted for sex, urinary creatinine, estimated glomerular filtration rate (eGFR), body mass index (BMI), smoking and exercise habits. Protein levels were measured on a log2-scale and transformed to a standard deviation (SD)-scale. The results are presented as Beta value, low (Cilow) and high (Cihigh) confidence intervals, *p*-values. FDR significance is given for each level of adjustment. The order is sorted on *p*-value. For comparison the molecular weight for each biomarker is presented in the right-hand column.

Adjustment for Sex, Urinary Creatinine, eGFR, BMI, Smoking and Exercise Habits					
Variable	Beta	Cilow	Cihigh	*p*-Value	MW (kDa)
Thrombospondin 2	0.66253	0.43343	0.89163	2.47 × 10^−7^ *	130
Interleukin 6	0.56538	0.36327	0.76749	5.26 × 10^−7^ *	23.7
Interleukin 8	0.4989	0.27203	0.72578	0.0000483 *	11.1
Hepatocyte growth factor	0.32414	0.16972	0.47856	0.0000976 *	83.1
Matrix metalloproteinase-12 (MMP-12)	0.51085	0.26328	0.75841	0.0001248 *	54
C-X-C motif chemokine 9	0.399	0.19804	0.59995	0.0002121 *	14
Osteoprotegerin	0.38552	0.17953	0.5915	0.0004508 *	46
Growth-regulated alpha protein	0.40438	0.18537	0.6234	0.0005307 *	11.3
C-X-C motif chemokine 6	0.39919	0.1622	0.63618	0.0014672 *	11.9
Oncostatin-M (OSM)	0.3936	0.1494	0.6378	0.0022716 *	28.5
Fatty acid-binding protein, intestinal	0.35665	0.13527	0.57804	0.002282 *	15.2
Protein S100-A12 (EN-RAGE)	0.36311	0.10689	0.61933	0.0068934 *	10.6
Spondin-2 (SPON2)	0.19605	0.05334	0.33877	0.008721	35.8
Polymeric immunoglobulin receptor (PIgR)	0.2687	0.06465	0.47276	0.011777	83.3
Proheparin-binding EGF-like growth factor (HB-EGF)	−0.30206	−0.56596	−0.03816	0.027785	23.7
Heme oxygenase 1 (HO-1)	0.21941	0.02154	0.41729	0.032871	32.8
C-C motif chemokine 4 (CCL4)	0.25296	0.02471	0.48122	0.032964	10.2
Latency-associated peptide transforming growth factor beta-1 (LAP TGF-beta-1)	0.28651	0.01783	0.5552	0.03996	44.3
Matrix metalloproteinase-7 (MMP-7)	0.22505	0.01171	0.43839	0.042082	29.7
Dickkopf-related protein 1 (Dkk-1)	0.24117	0.01231	0.47003	0.042289	28.7
Stem cell factor (SCF)	0.23776	0.00863	0.4669	0.045463	30.9
Urokinase-type plasminogen activator (uPA)	0.23353	0.00506	0.462	0.0487	48.5
Monocyte chemotactic protein 1 (MCP-1)	0.15326	−0.00317	0.30969	0.058581	11
Decorin (DCN)	−0.1959	−0.40101	0.00921	0.065049	39.7
Tissue factor (TF)	0.20699	−0.01075	0.42474	0.066297	33.1
Prolargin (PRELP)	0.16354	−0.01059	0.33766	0.069549	43.8
Delta and Notch-like epidermal growth factor-related receptor (DNER)	0.0687	−0.00519	0.14259	0.072327	78.5
Tumor necrosis factor (Ligand) superfamily. member 12 (TWEAK)	0.24865	−0.01961	0.5169	0.073198	27.2
Programmed cell death 1 ligand 2 (PD-L2)	0.19334	−0.02281	0.40949	0.083611	31
Kidney Injury Molecule (KIM1)	0.14807	−0.02614	0.32228	0.099858	38.7
Proteinase-activated receptor 1 (PAR-1)	−0.17143	−0.38624	0.04338	0.121937	47.4
V-set and immunoglobulin domain-containing protein 2 (VSIG2)	0.17382	−0.04422	0.39186	0.122328	34.3
Leukemia inhibitory factor (LIF)	0.18743	−0.06395	0.43881	0.14804	22
Tumor necrosis factor receptor superfamily member 9 (TNFRSF9)	0.15277	−0.05275	0.35829	0.149244	27.9
Protein AMBP (AMBP)	0.17428	−0.0668	0.41537	0.160594	39
T cell surface glycoprotein CD6 isoform (CD6)	0.13314	−0.05205	0.31833	0.162884	71.8
Heat shock 27 kDa protein (HSP 27)	0.15679	−0.06131	0.37488	0.162911	22.8
Fractalkine (CX3CL1)	0.08507	−0.0437	0.21385	0.199288	42.2
TNF-related apoptosis-inducing ligand receptor 2 (TRAIL-R2)	0.11637	−0.06322	0.29597	0.207955	47.9
CD40 ligand (CD40-L)	0.06835	−0.0419	0.1786	0.228085	29.3
CUB domain-containing protein 1 (CDCP1)	0.1487	−0.10124	0.39864	0.247213	92.9
Angiotensin-converting enzyme 2 (ACE2)	0.10271	−0.08248	0.2879	0.280465	92.5
Monocyte chemotactic protein 2 (MCP-2)	0.11316	−0.10873	0.33505	0.320708	11.2
Lipoprotein lipase (LPL)	0.12793	−0.13163	0.38749	0.337081	53.2
Tumor necrosis factor receptor superfamily member 10A (TNFRSF10A)	0.1034	−0.10735	0.31416	0.339281	50.1
Vascular endothelial growth factor A (VEGF-A)	0.10938	−0.11773	0.33648	0.348193	27
Prostasin (PRSS8)	0.11391	−0.12454	0.35237	0.352084	42.8
Interleukin-1 alpha (IL-1 alpha)	0.09469	−0.10705	0.29644	0.360489	30.6
Interleukin-17D (IL-17D)	−0.12737	−0.40046	0.14572	0.363538	21.9
Osteoclast-associated immunoglobulin-like receptor (hOSCAR)	−0.09501	−0.30164	0.11162	0.370336	30.5
Natural killer cell receptor 2B4 (CD244)	−0.106	−0.34078	0.12878	0.378981	41.6
Interleukin-18 (IL-18)	0.1087	−0.14085	0.35824	0.395949	22.3
T-cell surface glycoprotein CD5 (CD5)	0.07028	−0.0947	0.23525	0.406376	54.6
Interleukin-1 receptor antagonist protein (IL-1ra)	0.07691	−0.10753	0.26136	0.416301	20.1
Lactoylglutathione lyase (GLO1)	0.10629	−0.14975	0.36233	0.418381	20.8
Serine protease 27 (PRSS27)	0.09235	−0.14179	0.32649	0.44188	31.9
SLAM family member 5 (CD84)	0.07309	−0.12097	0.26715	0.462668	38.8
Interleukin-10 receptor subunit beta (IL-10RB)	−0.0726	−0.26956	0.12435	0.472201	37
Leukemia inhibitory factor receptor (LIF-R)	0.07275	−0.13932	0.28483	0.503391	123.7
Gastric intrinsic factor (GIF)	0.08151	−0.1627	0.32572	0.514985	45.4
Programmed cell death 1 ligand 1 (PD-L1)	0.04702	−0.09406	0.18811	0.51555	33.3
Galectin-9 (Gal-9)	−0.08683	−0.36148	0.18782	0.537352	39.5
Agouti-related protein (AGRP)	−0.05759	−0.24091	0.12574	0.539954	14.4
Cathepsin L1 (CTSL1)	−0.05031	−0.23928	0.13867	0.603348	37.6
Interleukin-1 receptor-like 2 (IL1RL2)	−0.03832	−0.18251	0.10587	0.603952	65.4
Adrenomedullin (ADM)	−0.04061	−0.21214	0.13092	0.643965	20.4
P-selectin glycoprotein ligand 1 (PSGL-1)	−0.03572	−0.19221	0.12078	0.655922	43.2
Lectin-like oxidized LDL receptor 1 (LOX-1)	0.05033	−0.17039	0.27105	0.656208	31
Carcinoembryonic antigen-related cell adhesion molecule 8 (CEACAM8)	0.03157	−0.11478	0.17792	0.673617	38.2
Transforming growth factor alpha (TGF-alpha)	0.05621	−0.20611	0.31854	0.675672	17
Alpha-L-iduronidase (IDUA)	−0.04533	−0.29791	0.20724	0.725979	72.7
Chymotrypsin C (CTRC)	−0.04294	−0.2859	0.20002	0.730005	29.5
Cystatin D (CST5)	−0.03128	−0.22667	0.16411	0.75455	16.1
Tumor necrosis factor receptor superfamily member 11A (TNFRSF11A)	0.02307	−0.13903	0.18517	0.781033	66
Interleukin-18 receptor 1 (IL-18R1)	0.01487	−0.11934	0.14908	0.828691	62.3
Placenta growth factor (PGF)	0.01696	−0.1476	0.18152	0.840446	24.8
Thrombomodulin (TM)	0.00998	−0.10482	0.12479	0.865108	60.3
Receptor for advanced glycosylation end products (RAGE)	−0.01822	−0.23653	0.2001	0.870512	42.8
Fms-related tyrosine kinase 3 ligand (Flt3L)	0.01844	−0.2344	0.27129	0.886689	26.4

* = Significant after adjustment for multiplicity testing using the false discovery rate (FDR).

## Data Availability

The datasets used and/or analyzed during the current study are available from the corresponding author on request.

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
