# Peer review of "Albumin Urinary Excretion Is Associated with Increased Levels of Urinary Chemokines, Cytokines, and Growth Factors Levels in Humans"

_biomolecules, 2021, doi:10.3390/biom11030396_

Round 1

Reviewer 1 Report

In the manuscript, the authors are interested in the association between albuminuria and urinary chemokines, cytokines and growth factors in the cohort of elderly participants. The results presented in this study are very interesting, useful and can have an important clinical message and practical implications. As well, up-date methods were used. Nevertheless, the manuscript suffers from some defects and is lacking in many aspects: - In general, the whole manuscript needs to be substantially expanded and filled in a number of missing items and comments. The individual sections are too brief and do not contain the necessary details. - In method section, the methods of statistical evaluation should be described in more detail. - In result section, in my opinion, both tables should be formatted with a classic border for better clarity in the text - I mean external framing. Individual parts of the results should be more and better described and interpreted. In table 2, it is not clear, what are „std“. - In particular, the authors should significantly expand and complement the discussion section. How these results could be interpreted from previous studies? The authors should compare to the findings with other studies, especially previous proteomic studies. A very specific group of the elderly is used in the study, this should be discussed. How this can affect urinary secretion of measured cytokines, chemokines and growth factors and thus also the results of this study? - Further, albuminuria is not only marker of renal dysfunctions but it is also an important risk factor for cardiovascular disease. This should be also discussed and emphasized. - Last but not least, what is the possible practical interpretation of the results and conclusions of this study? Some limitations of this study should be described and discussed in more detail at the end of the discussion section. - There are many abbreviations in whole manuscript, therefore I encourage authors add the list of abbreviation. In addition, some abbreviations are not explain at all.

Author Response

Dear Editor

We would like to than you and the reviewers for the valuable comments.

We have revised the manuscript according to these comments and below is a point by point response to the comments.

Yours sincerely

Anders Larsson

Comments and Suggestions for Authors

In the manuscript, the authors are interested in the association between albuminuria and urinary chemokines, cytokines and growth factors in the cohort of elderly participants. The results presented in this study are very interesting, useful and can have an important clinical message and practical implications. As well, up-date methods were used. Nevertheless, the manuscript suffers from some defects and is lacking in many aspects: -

In general, the whole manuscript needs to be substantially expanded and filled in a number of missing items and comments.

In method section, the methods of statistical evaluation should be described in more detail.

The methods section has now been expanded to include more details

In result section, in my opinion, both tables should be formatted with a classic border for better clarity in the text - I mean external framing.   External framing has been added.

Individual parts of the results should be more and better described and interpreted. In table 2, it is not clear, what are „std“. –  The table has been updated, std has been removed and the abbreviations has been replaced by the full names.

In particular, the authors should significantly expand and complement the discussion section. How these results could be interpreted from previous studies? The authors should compare to the findings with other studies, especially previous proteomic studies. A very specific group of the elderly is used in the study, this should be discussed. How this can affect urinary secretion of measured cytokines, chemokines and growth factors and thus also the results of this study? –  

The study focused on the measurements in urine.  There were no previous studies made in people with preserved kidney function to compare with. The aim was exploratory to find out if any of these markers do appear in urine from apparently healthy people.  Another aim was to investigate if these proteomics markers behaved like albumin or not as an indication of the source of the cytokines. The results indicate that the urine cytokine levels were not due to filtration in the glomeruli.

The study subjects were elderly individuals, probably suffering from some degree of age-related renal dysfunction, including to some extent nephrosclerosis. However, they do not report any specific renal disease. Being an elderly group of individuals, they are rep-resentative for populations that most frequently seek health care.

Further, albuminuria is not only marker of renal dysfunctions but it is also an important risk factor for cardiovascular disease.

Albuminuria is not only used as a marker of renal dysfunctions but it is also a well-recognized predictor of cardiovascular risk in both community-based populations and high-risk populations. Proinflammatory cytokines have in several studies been shown to be cardiovascular risk markers. Acute or chronic injury to the kidneys or heart is known to be able to induce acute or chronic dysfunction of the other organ. These interactions are often referred to as the cardiorenal syndrome. The mechanisms behind these interactions are poorly understood, but could be mediated by cytokines. Several cytokines have been reported to be cardiovascular risk markers.

This should be also discussed and emphasized. - Last but not least, what is the possible practical interpretation of the results and conclusions of this study?  Apparently healthy people do excrete multiple biomarkers captured with this proteomics platform, in addition to albumin.  These results may serve as a reference when studying patients with specific renal diseases.

Some limitations of this study should be described and discussed in more detail at the end of the discussion section. –   

Added to the discussion: Some limitations of the study should be noted, such as a target on 75 year old indi-viduals who may have a certain degree of nephrosclerosis and a slightly reduced renal function. Urinary albumin is not a pure and exclusive marker of increased glomerular permeability, since some albumin is filtered in glomeruli and subsequently reabsorbed in the proximal tubule.

There are many abbreviations in whole manuscript, therefore I encourage authors add the list of abbreviation. In addition, some abbreviations are not explain at all.  

We have replaced the abbreviations with the full names.

Reviewer 2 Report

For this otherwise interesting study, I have nevertheless quite a lot of remarks:

  1. This paper suffers from poor English in many sentences, e.g. verb in singular, with the subject in plural, or vice versa: to be corrected.
  2. Abstract, line 27: “….both small and large urine proteins that are linked to inflammation. These associations are thus not merely caused by a leakage of large plasma proteins due to glomerular damage”. The conclusion is justified, but then exactly what form of “inflammation” did the authors expect to find in their 70-years old “healthy” study population?
  3. Methods, line 72: “A total of 90 individuals without CVD, diabetes, or stroke were included”. Thus, a healthy study population. The authors should more clearly expand how the described relationship between the discussed “significative” urinary proteins and urinary albumin in a HEALTHY study population should necessarily (?) become even more significative in a non-healthy study population, as given in the Discussion.
  4. Table 1: if the mean Urine albumin of 16.5 mg/L given in table 1, seems indeed within normal limits, the authors should at least mention these norms somewhere in the text or in Table 1, which is now not the case.
  5. Table 2, giving the overview of the 79 obtained results indeed, but giving nowhere an explication of the many used abbreviations. If an electronic appendix with full explications of all 79 biomarkers is technically impossible or not considered imperative, then at least explications of the used abbreviations are needed-for some of these, in the text of the Discussion
  6. Table 2: “Relationships between 79 urinary proteins and urinary albumin adjusted for sex, urinary creatinine, eGFR, BMI, smoking, and exercise habits”. In the Methods section 2.4. Statistics, it is not clear (at least for a non-statistician) exactly what kind of correlation or “relationship” was calculated, the result of which is now given in Table 2. Apparently, the studied proteins were all “correlated” to the albuminuria, expressed in spot samples as mg/L, and NOT as the corresponding associations with the albumin-creatinine ratio (ACR) (line 153)?
  7. It is not clear neither exactly why “as expected” (line 151) this association of albumin in mg/L proper, was indeed stronger than the corresponding associations with the albumin-creatinine ratio (ACR). This ACR is always preferable for evaluation of the true degree of albuminuria, since corrected for diminished GFR (as in the here used study group, with “slightly reduced renal function”, line 188), and consequently for diminished urinary creatinine excretion. If ACR is in clinical practice better indeed for evaluation of the true degree of albuminuria, it should be also the case for evaluation of the urinary excretion of other “inflammatory, etc “ proteins?

Author Response

Dear Editor

We would like to than you and the reviewers for the valuable comments.

We have revised the manuscript according to these comments and below is a point by point response to the comments.

Yours sincerely

Anders Larsson

Reviewer 2

17 Jan 2021 23:23:10

  1. This paper suffers from poor English in many sentences, e.g. verb in singular, with the subject in plural, or vice versa: to be corrected. We have revised the manuscript to improve the English.
  2. Abstract, line 27: “….both small and large urine proteins that are linked to inflammation. These associations are thus not merely caused by a leakage of large plasma proteins due to glomerular damage”. The conclusion is justified, but then exactly what form of “inflammation” did the authors expect to find in their 70-years old “healthy” study population?  This is an exploratory study. We assumed that increased urine albumin would be caused by glomerular damage. Organ damage is usually associated with an inflammatory response of varying degree.
  3. Methods, line 72: “A total of 90 individuals without CVD, diabetes, or stroke were included”. Thus, a healthy study population. The authors should more clearly expand how the described relationship between the discussed “significative” urinary proteins and urinary albumin in a HEALTHY study population should necessarily (?) become even more significative in a non-healthy study population, as given in the Discussion. In a healthy population the urine albumin concentration varies less than in a patient population with severe kidney disease. A larger distribution range for one of the markers are usually associated with stronger significance values. We have however no data in this study to support this statement so we have removed it in the revised version.
  4. Table 1: if the mean Urine albumin of 16.5 mg/L given in table 1, seems indeed within normal limits, the authors should at least mention these norms somewhere in the text or in Table 1, which is now not the case. Added to the Materials section: A urine albumin of > 20 mg/L or a urine albumin/creatinine ratio of > 2.5 mg/mmol is suggestive of microalbuminuria.
  5. Table 2, giving the overview of the 79 obtained results indeed, but giving nowhere an explication of the many used abbreviations. If an electronic appendix with full explications of all 79 biomarkers is technically impossible or not considered imperative, then at least explications of the used abbreviations are needed-for some of these, in the text of the Discussion  We have revised Table 2 and replaced the abbreviations with the full names.
  6. Table 2: “Relationships between 79 urinary proteins and urinary albumin adjusted for sex, urinary creatinine, eGFR, BMI, smoking, and exercise habits”. In the Methods section 2.4. Statistics, it is not clear (at least for a non-statistician) exactly what kind of correlation or “relationship” was calculated, the result of which is now given in Table 2. Apparently, the studied proteins were all “correlated” to the albuminuria, expressed in spot samples as mg/L, and NOT as the corresponding associations with the albumin-creatinine ratio (ACR) (line 153)? The statistical description has been expanded in the revised manuscript.

  1. It is not clear neither exactly why “as expected” (line 151) this association of albumin in mg/L proper, was indeed stronger than the corresponding associations with the albumin-creatinine ratio (ACR). This ACR is always preferable for evaluation of the true degree of albuminuria, since corrected for diminished GFR (as in the here used study group, with “slightly reduced renal function”, line 188), and consequently for diminished urinary creatinine excretion. If ACR is in clinical practice better indeed for evaluation of the true degree of albuminuria, it should be also the case for evaluation of the urinary excretion of other “inflammatory, etc “ proteins?  

You are absolutely right that in clinical practice ACR is very useful, especially when following an individual patient with a stable muscle mass and GFR over time.  However, in this case PEA values are a concentration measure and it is preferrable to compare two concentration measures or two creatinine adjusted measurements. Comparing one concentration measure with one creatinine adjusted measurement will increase the variation. We have now omitted this sentence.

Round 2

Reviewer 1 Report

The authors have made the required changes and recommendations, the manuscript can be accepted to the publication.

Author Response

Thanks for your reoprt!

Reviewer 2 Report

This MS is still very questionable in some aspects, despite the indeed “ameliorated” R1 form. The authors should, at least in their Discussion, and consequently also in their still now unchanged Abstract, more clearly stress and repeat that their findings apply so far only to a NORMAL population, and that extrapolations of their findings to PATHOLOGICAL situations is not always quite obvious. In following remarks, I will try to give some examples:

  1. the very first word of the (unchanged) title is not appropriate, since in common clinical use, “Albuminuria” denotes a renal affliction, not a normal renal finding. According to your own definitions on lines 81 & 82 (BTW, not given in the first version!), a urinary albumin of > 20 mg/L is indeed abnormal or “suggestive of microalbuminuria”. The current mean urinary albumin of 16.5 mg/L (Table 1) consequently is NORMAL, and not even a low degree of microalbuminuria. Please prefer in the title& text the more diplomatic term “albumin urinary excretion”.
  2. Abstract: line 27: “ urine proteins linked to inflammation”. Again, yes these named urine proteins might be (or will be later) linked to inflammation, but the problem in this study remains: to what form of so-called inflammation did the authors link the current increased urinary proteins? One cannot claim a link in the abstract, if this described link is not proven in the subsequent text .The now only described link (Discussion, lines 151-3) is to a vague “some degree of age-related renal dysfunction, including TO SOME EXTENT nephrosclerosis”. The latter (nephrosclerosis) is thus purely hypothetical, and not even a true ongoing inflammation.
  3. Abstract: line 27-28: “ not merely caused by a leakage of large plasma proteins due to glomerular damage”. Again, this bold statement in the abstract, is not proven in the subsequent text. If large proteins, e.g. IgG globulins, are found back in the urine (= definition of a-selective proteinuria) in some serious renal diseases, by which other mechanism do the authors presume then that this can happen, except for an obvious leakage of large plasma proteins due to (acute or chronic) glomerular damage?
  4. Methods, line 67-69: stating that the study subjects were chosen “from 70 years old”, but then that “urine collections were collected at 75 years” is confusing for the reader. Better stick to 75years, in accordance with Table 1.
  5. Discussion, lines 161-162: “…increase the urine concentration of large proteins, while the small ones will remain unchanged”. This other bold statement seems paradoxical, and consequently should be confirmed at least by (some) data of nephrological studies.
  6. Discussion, lines 178-9: “or not, as an indication of the source of the cytokines”. This sentence remains incomprehensible, and should be re-phrased.  

Author Response

This MS is still very questionable in some aspects, despite the indeed “ameliorated” R1 form. The authors should, at least in their Discussion, and consequently also in their still now unchanged Abstract, more clearly stress and repeat that their findings apply so far only to a NORMAL population, and that extrapolations of their findings to PATHOLOGICAL situations is not always quite obvious. In following remarks, following remarks, I will try to give some examples:

Response: We have revised manuscript and clearly states that the study population is a normal population in the abstract and discussion.

  1. the very first word of the (unchanged) title is not appropriate, since in common clinical use, “Albuminuria” denotes a renal affliction, not a normal renal finding. According to your own definitions on lines 81 & 82 (BTW, not given in the first version!), a urinary albumin of > 20 mg/L is indeed abnormal or “suggestive of microalbuminuria”. The current mean urinary albumin of 16.5 mg/L (Table 1) consequently is NORMAL, and not even a low degree of microalbuminuria. Please prefer in the title& text the more diplomatic term “albumin urinary excretion”.

Changed as recommended

  1. Abstract: line 27: “ urine proteins linked to inflammation”. Again, yes these named urine proteins might be (or will be later) linked to inflammation, but the problem in this study remains: to what form of so-called inflammation did the authors link the current increased urinary proteins? One cannot claim a link in the abstract, if this described link is not proven in the subsequent text .The now only described link (Discussion, lines 151-3) is to a vague “some degree of age-related renal dysfunction, including TO SOME EXTENT nephrosclerosis”. The latter (nephrosclerosis) is thus purely hypothetical, and not even a true ongoing inflammation.

The cytokines used in the study are from an inflammatory panel. The link that we stated was between the markers in the proseek panels and inflammation. A pubmed search on IL6 and inflammation resulted in 53000 hits. We thus believe that the roles of these cytokines are fairly well established. Maybe we interpret the sentence differently. However, the association with inflammation is not essential for the content of the manuscript so we changed the abstract to:  In this study we found associations between urinary albumin and both small and large urine proteins.

  1. Abstract: line 27-28: “ not merely caused by a leakage of large plasma proteins due to glomerular damage”. Again, this bold statement in the abstract, is not proven in the subsequent text. If large proteins, e.g. IgG globulins, are found back in the urine (= definition of a-selective proteinuria) in some serious renal diseases, by which other mechanism do the authors presume then that this can happen, except for an obvious leakage of large plasma proteins due to (acute or chronic) glomerular damage?

The sentence has been omitted in the revised version.

  1. Methods, line 67-69: stating that the study subjects were chosen “from 70 years old”, but then that “urine collections were collected at 75 years” is confusing for the reader. Better stick to 75years, in accordance with Table 1.

Changed to: The subjects were chosen from 75 year old persons living in the community of Uppsala, Sweden.

  1. Discussion, lines 161-162: “…increase the urine concentration of large proteins, while the small ones will remain unchanged”. This other bold statement seems paradoxical, and consequently should be confirmed at least by (some) data of nephrological studies.

The sentence has been omitted in the revised version.

  1. Discussion, lines 178-9: “or not, as an indication of the source of the cytokines”. This sentence remains incomprehensible, and should be re-phrased.  

The sentence has been omitted in the revised version.